# Progress in Fermented Unconventional Feed Application in Monogastric Animal Production in China

**Haoxuan Sun** [1,2], **Xinyue Kang** [3], **Huize Tan** [1,2], **Huiyi Cai** [4] **and Dan Chen** [1,2,*]

1  WENS Foodstuff Group Co., Ltd., Yunfu 527400, China
2  Key Laboratory of Animal Nutrition and Feed Science, Ministry of Agriculture, Yunfu 527400, China
3  College of Life Science and Technology, Southwest Minzu University, Chengdu 610041, China
4  National Engineering Research Center of Biological Feed Development, Beijing 100044, China
*  Correspondence: 82101215368@caas.cn

**Abstract:** Unconventional animal feeds present distinct features and considerable variations. However, their efficacy in monogastric animals is hindered by high levels of anti-nutritional elements and subpar palatability. Feed fermentation could offer a solution to these issues. Moreover, fermented unconventional feeds deliver notable economic advantages and represent a viable alternative to antibiotic growth promoters, particularly in the context of antibiotic restrictions, promising considerable potential. This review provides an in-depth exploration of the types, characteristics, fermentation processes, application outcomes, associated challenges, and prospects of fermented unconventional feeds in monogastric animals. We anticipate that this comprehensive overview will serve as a valuable reference for developing and utilizing unconventional feed resources in the feed industry.

**Keywords:** fermentation; microorganisms; animal feeding ingredients; intestinal health; growth performance; animal nutrition

## 1. Introduction

In contemporary China, the enforcement of regulations prohibiting the use of antibiotics presents a significant challenge to the livestock industry. Managing this challenge is a top priority for all livestock professionals. One promising solution is animal feed fermentation. Probiotics and their metabolites in fermented feed play an important role in reducing or replacing antibiotics [1]. Fermentation offers multiple advantages, including enhanced feed palatability, removal of anti-nutritional factors, improved animal gut health, and superior meat quality [2]. Fermentation is broadly defined as the process of utilizing microbial activity to produce metabolic products [3]. The practice of fermentation dates back thousands of years, when humans first began fermenting products using yeast. In the early 20th century, the Finnish biochemist Artturi Ilmari Virtanen conducted groundbreaking research on silage feed fermentation [4]. Virtanen developed a method based on this principle that prevented the feed from spoiling and maintained its usability and nutritional value. Virtanen was awarded the Nobel Prize in Chemistry in 1945 for his pioneering work.

Silage is mainly used to feed ruminants [5]; however, monogastric animal production now plays an important role in human life. Monogastric animals mainly include poultry and pigs. People feed monogastric animals to obtain more meat and eggs to meet people's nutritional needs. Feed is a necessary nutrient for animals. The quality of feed determines the quality of animal products [6]. Therefore, how to improve feed quality is a compulsory subject for every researcher in the feed industry. As science and technology progressed, various disciplines, such as fermentation, genetics, and enzymatic engineering, saw continuous improvement. People gradually shifted their focus towards applying fermentation technology to animal feed. This evolution began with silage feed fermentation, advanced to fermentation of individual feed ingredients, and more recently expanded to

fermentation of complete diets [7]. Fermentation, as a tool for enhancing the quality of livestock products, could promote animal intestinal health, improve growth performance, and improve immune function. Therefore, fermented feed has garnered increasing attention from researchers [8]. In 2021, the Livestock and Veterinary Bureau of the Chinese Ministry of Agriculture and Rural Affairs initiated the "Program for Reducing Corn and Soybean Meal in Feed". This program aims to promote the adoption of low-protein diets and suggests substituting soybean meal with unconventional feed ingredients, such as sorghum, barley, and oats, to reduce China's long-standing dependence on soybean meal imports. However, many unconventional feed ingredients are characterized by poor palatability and high levels of anti-nutritional factors, which limit their effectiveness. Although most monogastric animals, including pigs and poultry, are characterized by strong enzymatic digestion and weak microbial digestion, differences among them are still noted [9]. Pigs have teeth, while poultry do not, so pigs can chew their feed. Unlike pigs, poultry have glandular and muscular stomachs, and their crop functions to soften the feed [10]. Poultry are small, and their intestines are short, so the feed transit time through the digestion tract is short. Furthermore, poultry's enzymatic and microbial fermentation digestion is weaker than in pigs [11]. Unlike conventional feed, such as soybean meal, monogastric animals have difficulty digesting diets with high crude fiber content, such as unconventional feeds, with their available enzymes, resulting in poor nutrient utilization. Fermentation of these feed ingredients could help overcome these shortcomings. This article overviews the meager research on unconventional feed resource fermentation and its use in monogastric animals. Our objective was to offer theoretical support for developing unconventional feed resources.

## 2. Classification of Fermented Unconventional Feeds

Based on international feed classification standards, feeds are typically categorized into eight classes, including protein feed, energy feed, roughage, green forage, silage, mineral feed, vitamin feed, and feed additives. Unconventional feeds are also classified using these international standards. It is important to note that feed additives do not require fermentation, and roughage fermentation is primarily applied in ruminant animal husbandry. Additionally, mineral and vitamin feed fermentation might result in nutrient losses [12]. Consequently, this review will focus solely on protein and energy feed fermentation and advancements in the study of silage feeds.

### 2.1. Fermentation of Protein Feeds

Protein feeds are defined as feeds with a natural moisture content of less than 45%, a crude fiber content in dry matter of less than 18%, and a crude protein content of at least 20% [10]. These feeds can be further categorized into four subgroups based on their source: plant-based protein feeds, animal-based protein feeds, non-protein nitrogen feeds, and single-cell protein feeds. Compared with conventional feeds such as soybean meal, unconventional protein feeds have poor palatability, uneven nutritional composition, and low nutritional value, and they contain a variety of anti-nutritional factors and toxicants. Fermentation could effectively solve these problems. It is widely recognized that the nutritional characteristics of fermented feeds depend on various factors, including the selection of microorganisms, substrates, and fermentation conditions (e.g., temperature and duration). In this review, we have compiled information on these aspects and their use in the fermentation of unconventional protein feeds, as summarized in Table 1. Numerous studies have demonstrated the nutritional improvements achieved through protein feed fermentation. These improvements include increased crude protein content and decreased crude fiber content [13]. Furthermore, such fermentation has been shown to mitigate the adverse effects of heat stress in broilers [14]. The underlying reason for these improvements lies in the ability of fermentation to break down large molecular nutrients in protein feeds into smaller molecules, such as peptides, free amino acids, and oligosaccharides. This degradation improves their digestibility and absorption, as illustrated in Figure 1.

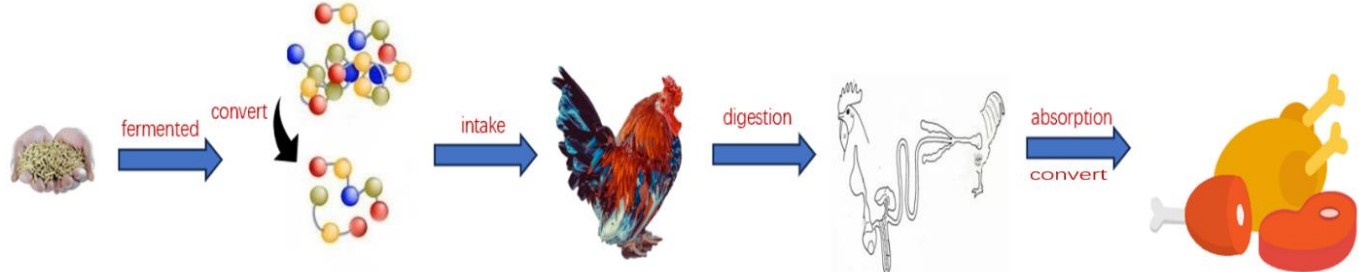

**Figure 1.** Feed fermentation converts macromolecular substances into small molecular substances, promoting absorption and transformation.

**Table 1.** Microorganisms, Substrates, and Process Conditions for Fermentation of Unconventional Protein Feeds.

| Substrate | Microorganisms | Process Conditions | Reference |
|---|---|---|---|
| Cottonseed meal | *Cellulosimicrobium funkei* | Inoculum size, 10%; material-to-water ratio, 1:0.5; temperature, 35 °C; fermentation time, 144 h | [15] |
| Peanut meal | *Streptococcus thermophilus* and *Lactobacillus delbrueckii* subsp. *bulgaricus* | Anaerobic fermentation at 37 °C for 48 h | [16] |
| Rapeseed meal | *Lactobacillus acidophilus*, *Bacillus subtilis*, and *Saccharomyces cerevisiae* | Temperature, 33 °C; material-to-water ratio, 1:1; fermentation time, 84 h; inoculum size, 6% | [17] |
| Cottonseed meal | *Bacillus* sp. *strains* | Anaerobic fermentation at room temperature for 14 days | [18] |
| Flaxseed cake | *Aspergillus niger* and *Candida utilis* | Temperature, 30 °C; fermentation time, 72 h | [19] |

### 2.2. Fermentation of Energy Feeds

Energy feeds are characterized by a crude fiber content of less than 18% and a crude protein content lower than 20% in dry matter. In China, conventional energy feeds include maize and wheat, while unconventional ones include grains like sorghum and oats (the primary source), tuber and root crops such as sweet potatoes, and by-products like vinegar production residue, distillers' grains, and beet pulp. Grains are rich in starch but have low crude fiber content, tend to be low in protein, have low overall quality, and exhibit unbalanced mineral and vitamin content. Tuber and root crops have high moisture content and are abundant in non-nitrogenous extracts. By-product feeds have several drawbacks, including high fiber content, elevated moisture levels, significant anti-nutritional factors, and susceptibility to spoilage. These factors make them less commonly used in practical applications for monogastric animals. Nevertheless, unconventional energy feeds can undergo significant improvements through microbial fermentation [17]. This process results in a notable increase in crude protein and organic acid content, leading to enhanced palatability and nutritional value [19]. Furthermore, fermentation helps reduce the presence of anti-nutritional factors, effectively lowering the overall cost of feed production. Table 2 presents information about the microorganisms, substrates, and process conditions employed in the fermentation of unconventional energy feeds.

**Table 2.** Microorganisms, Substrates, and Process Conditions for Fermentation of Unconventional Energy Feeds.

| Substrate | Microorganisms | Process Conditions | Reference |
|---|---|---|---|
| *Ginkgo biloba* kernel juice | *Lactobacillus plantarum* | Fermented at 37 °C for 48 h | [20] |
| *Ginkgo biloba* leaves | *Candida utilis* and *Aspergillus niger* | Cultivated at 28–30 °C for 48 h | [21] |
| *Flammulina velutipes* by-products | *Lactobacillus plantarum* and *Saccharomyces cerevisae* | 0.1% probiotics at 40 °C for 24 h | [22] |
| Vegetable waste (kale) | *Lactobacillus plantarum* | Fermented at 25–30 °C for 10–15 days with a humidity of 65–75%. | [23] |

*2.3. Silage Feed*

As mentioned above, silage feed is a valuable option in animal nutrition. It involves green forage fermentation using lactic acid bacteria under anaerobic, low-pH conditions. This process serves two main purposes: adjusting the nutritional composition of the forage and preserving it for later use. Silage feed is transformed during fermentation. Its crude fiber content decreases, it acquires an aromatic odor, becomes more palatable, and, importantly, the lactic acid produced during this process inhibits the growth of harmful bacteria and extends its shelf life. Silage feed has a low crude fiber content and a high nutritional value, making it suitable for monogastric animals when included in their diets in appropriate proportions. Some studies have shown that adding apple pomace-mixed silage to the basal diet could improve the feed conversion efficiency of finishing pigs [24]. Many studies have reported on the improvement in monogastric animal meat quality due to silage feed [25].

Silage feed quality can be evaluated using various indicators. One common measure is the lactic acid to acetic acid ratio, which is typically 2.5–3.0 in high-quality silage feed. The silage feed odor is another criterion for assessing its quality. Well-fermented silage feed should not have a strong or peculiar odor. Lactic acid is the main organic acid produced during fermentation and is nearly odorless; however, some mild odor might be present since acetic acid is the second most abundant organic acid formed during fermentation [5]. However, it is worth noting that the quality of silage feed can vary depending on the growth stage at which the forage was harvested for silage production. The choice of growth stage or harvest period significantly impacts the nutritional value and fermentation quality of silage feed [25].

Silage processing can substantially improve the nutritional value of the feed. For example, *Pennisetum giganteum*, a grass species commonly grown in southern China, exhibits remarkable productivity. It can be harvested 6–8 times yearly and is resilient to flooding, drought, and high temperatures [26]. This grass yields approximately 254 tons of fresh grass per hectare, making it a high-yield option [27]. Given the region's rainy climate and seasonal surplus of *P. giganteum*, silage has emerged as the preferred processing method [28]. Research has demonstrated that the anaerobic ensilage of *P. giganteum* enhances the metabolism of fats, cofactors, vitamins, energy, and amino acids, thereby increasing its nutritional value [29].

**3. Application of Fermented Unconventional Feeds in Monogastric Animals**

*3.1. Growth Performance*

One of the primary objectives of adding fermented feeds to animal diets (including rabbits [30], poultry [31], geese [32], ducks [15], and pigs [33]) is to enhance growth performance. Previous research has consistently shown that feeding animals with fermented feeds positively impacted their growth, promoted weight gain [34], and improved feed conversion rates [32]. Xu and colleagues [35] conducted a meta-analysis focusing on the impact of fermented feeds on pig growth performance. They evaluated 3271 articles, retaining 30 from 2000 to 2019 (involving 3562 pigs) for the meta-analysis. Their results

indicated that fermented feeds could increase the daily weight gain and feed conversion efficiency of weaned piglets and finishing pigs. This positive effect was attributed to improved nutritional value and utilization of feeds gained during fermentation. The meta-analysis suggested that most analyzed studies supported its findings, so clearly numerous studies have investigated the impact of fermented unconventional feeds on the growth performance of monogastric animals, with a substantial body of research confirming these findings. Some instances of improved growth performance in monogastric animals fed fermented unconventional feeds are summarized in Table 3.

**Table 3.** Effects of Fermented Unconventional Feeds on the Growth Performance of Monogastric Animals.

| Substrate and Dosage | Animal and Stage | Growth Performance | Reference |
|---|---|---|---|
| Garlic powder (4 g/kg) | Growing-finishing pigs, 12 weeks | FCR significantly decreased from 2.96 to 2.61 compared to the control group | [36] |
| Bamboo fiber (4% substitution for wheat bran) | Pregnant sows, from day 80 of gestation to the end of lactation | Average daily feed intake significantly increased from 6.37 kg/day to 7.56 kg/day compared to the control group | [37] |
| Bamboo powder (5% substitution for wheat bran) | Growing-finishing pigs, a 75-day experiment | FCR decreased from 2.81 to 2.80 compared to the control group, with no adverse effects | [38] |
| Substrate composed of 80% rice distillers' grains and 20% wheat bran (8% substitution for corn) | Finishing pigs, a 50-day experiment | FCR significantly decreased from 3.03 to 2.98 compared to the control group | [39] |
| The base diet contaminated with 10% aflatoxin B1 fermented cottonseed meal (16.0 μg aflatoxin B1/kg) | Cherry Valley ducklings, a 14-day experiment | FCR significantly decreased from 2.19 to 1.92 compared to 10% aflatoxin B1 cottonseed meal (96.8 μg aflatoxin B1/kg) | [15] |
| Ginkgo biloba leaves (0.4%) | Broilers, a 42-day experiment in the grower phase | FCR significantly decreased from 1.75 to 1.62 compared to the control group | [21] |
| *Flammulina velutipes* by-products (70%) | Growing-fattening Berkshire pigs, aged approximately 112 days, fed them until reaching 105 kg | Feed efficiency significantly decreased from 0.305 g/g to 0.227 g/g compared to the control group | [22] |
| *Vegetable wastes* (kale, %) | Yellow chickens, 1–21 days old | FCR was significantly lower than the control group | [23] |
| Dandelion (1000 mg/kg addition) | Broilers, 1–42 days old | FCR decreased from 1.73 to 1.64 compared to the control group | [40] |
| Okara (55% substitution for corn, 72% substitution for soybean meal) | Growing pigs, 55 days | FCR significantly decreased from 3.06 to 2.89 compared to the control group | [41] |
| *Ginkgo biloba* leaves (4.5 g/kg) | Broiler Chickens, 1–42 days | FCR significantly decreased from 1.44 to 1.38 compared to the control group | [42] |
| Pomace-mixed silage | Finishing pigs, eight weeks | Feed efficiency (G/F) significantly increased from 0.29 g/g to 0.42 g/g compared to the control group | [24] |
| Herbal residues (5%) | Broilers, 1–42 days | FCR significantly decreased from 4.60 to 3.80 compared to the control group | [43] |
| Malic acid (8 g/kg) | Broilers, 1–21 days | FCR significantly decreased from 1.64 to 1.38 compared to the control group | [44] |
| Citri Sarcodactylis Fructus by-products (3%) | Broilers, 1–42 days | FCR significantly decreased from 4.60 to 3.37 compared to the control group | [45] |

The microorganisms in the fermentation process broke down complex carbohydrates and organic compounds in the feed that are otherwise difficult to digest. The metabolites produced, and the enzymes secreted during fermentation, enhance nutrient utilization in the feed. This effect is particularly pronounced in young animals since their digestive systems are underdeveloped. Supplementing their diets with fermented feeds could compensate for their immature digestive system. It is important to note that the impact of fermented feeds on growth performance can vary depending on the growth stage of the animals [40]. The effects on nutrient utilization might be mild in animals with a fully developed digestive system, such as broilers in the later growth stages [46]. This observation was supported by research conducted in broilers fed fermented feeds throughout their growth period, showing that the improvement in nutrient utilization was more pronounced in the early growth stages when the digestive system was still developing [47]. This difference is because the digestive system of young broilers is not fully developed during the starter phase, and their levels of endogenous digestive enzymes are relatively low. Bacteria could secrete digestive enzymes such as protease and amylase during fermentation. For example, secreted proteolytic enzymes degrade proteins into short peptides, thus promoting feed digestion and absorption in early-growing broilers. Furthermore, *L. plantarum* and *B. subtilis* increase the content of short peptides (<600 Da) in the feed by nearly 62% [48], and the proportion of short peptides increases with the duration of fermentation [49]. The higher the digestive enzyme activity in the intestine, the higher the feed utilization absorption rates [50].

### 3.2. Immune Function

Immune function is a paramount parameter of interest for animal nutritionists and producers. Previous research has demonstrated that fermented feed could enhance the immune function of animals [51], particularly in young animals who often have underdeveloped immune organs and weaker disease resistance [52]. For instance, early weaning in pigs could lead to weaning stress and result in diseases like diarrhea [53]. Adding fermented feeds to the diets of weaned piglets could promote their intestinal development, improve survival rates, enhance growth performance, and mitigate the effects of weaning stress [54].

Some instances of how fermented unconventional feeds affect the immune function of monogastric animals are presented in Table 4. Animal immune parameters are diverse and challenging to cover comprehensively [55]. Therefore, this table will focus on discussing immune organ indexes and immunoglobulins (IgA, IgM, and IgG) in monogastric animals. The primary reason for immune function enhancement in monogastric animals when fed fermented unconventional feeds is likely associated with the production of organic acids during the fermentation process. These organic acids can inhibit the growth of harmful bacteria in the digestive tract, the largest immune organ in the body. Additionally, fermentation could form small peptides, thus increasing the concentration of immunoglobulin in animal serum [56]. In addition, fermented feed probably promotes the immune system because probiotics can promote a balanced intestinal microflora and inhibit the colonization of the gastrointestinal tract by pathogens by promoting the production of antibodies, competition for attachment sites and nutrients, and bactericidal effects [57].

**Table 4.** Effects of Fermented Unconventional Feeds on the Immune Function of Monogastric Animals.

| Substrate and Dosage | Animal and Stage | Immune Function | Reference |
|---|---|---|---|
| Cottonseed meal (8% substitution for soybean meal) | Broilers, 1–42 days old | IgM significantly increased from 0.11 to 0.16 mg/mL compared to the control group | [58] |
| Rapeseed meal (15% addition) | Broilers, 1–42 days old | IgG increased from 10.58 to 11.11 mg/mL compared to the control group | [13] |

**Table 4.** *Cont.*

| Substrate and Dosage | Animal and Stage | Immune Function | Reference |
|---|---|---|---|
| Dandelion (500 mg/kg addition) | Broilers, 1–42 days old | Spleen indexes significantly increased from 0.08 to 0.13 compared to the control group | [40] |
| Rapeseed meal (15% substitution for soybean meal) | Broilers, 1–42 days old | IgG significantly increased from 0.2 mg/mL to 0.4 mg/mL compared to the control group | [59] |
| Garlic powder (4 g/kg) | Growing-finishing pigs, 12 weeks | IgG significantly increased from 1284 mg/mL to 1483 mg/mL compared to the control group | [36] |
| Bamboo powder (5% substitution for wheat bran) | Growing-finishing pigs, a 75-day experiment | IgA significantly increased from 0.97 g/L to 1.27 g/L compared to the control group | [38] |
| Substrate composed of 80% rice distillers' grains and 20% wheat bran (8% substitution for corn) | Finishing pigs, a 50-day experiment | IgA significantly increased from 0.59 g/L to 0.63 g/L compared to the control group | [39] |
| Malic acid (12 g/kg) | Broilers, 1–42 days | IgG significantly increased from 4.42 g/L to 4.48 g/L compared to the control group | [44] |

### 3.3. Animal Products

The primary objective of animal husbandry is to obtain various animal products, with a particular emphasis on achieving high-quality results—an everlasting pursuit for professionals in animal nutrition. These encompass a wide range of animal products, including meat, eggs, milk, fur, and more. In monogastric animals, such as those commonly raised in China, the predominant focus is meat and eggs, primarily the former [60]. Due to space constraints, this review will solely discuss meat quality-related issues.

Meat quality determination entails a comprehensive assessment of specific physical and chemical attributes that collectively define the appearance, palatability, and nutritional value of fresh and processed meat [61]. Typically, meat quality attributes are categorized into four main dimensions: hygiene and safety, nutritional content, sensory characteristics, and processing suitability [62]. Meat hygiene and safety evaluation encompasses the evaluation of microbial composition and quantity (indicative of meat spoilage and degradation) and the levels of antibiotics, hormones, and other residues [63]. Nutritional value considerations encompass protein and fat content, amino acid profiles, micronutrients, vitamins, and more [47]. The sensory attributes category includes factors such as flavor, tenderness, juiciness, drip loss, cooking yield, marbling, freshness, and color [64]. Notably, high-quality meat often exhibits low drip loss, indicative of excellent water-holding capacity—a crucial parameter for evaluating sensory quality [65]. Processing quality assessment includes aspects like oxidative stability [66].

Meat flavor comprises taste and aroma. Human perception is influenced by the taste and olfactory systems, which are affected by the physical structure and protein composition of the meat [67]. The meat's nutritional value hinges on the effective breakdown of its proteins into peptides and free amino acids by digestive enzymes for utilization within the human body [68]. Nutrition plays a pivotal role in shaping meat quality, with adequate nutrient provision being a prerequisite for high-quality meat products [69]. Nutrition-related factors are the most direct drivers of meat quality and offer the swiftest and most effective means to enact changes in meat quality within the current technological landscape [25].

This review exclusively delved into the impact of fermented unconventional feeds on the nutritional value and sensory attributes of meat. Some of these effects are outlined in Table 5. A meta-analysis showed that fermented feeds affected the meat quality and had a good promoting effect [35]. There are two popular explanations for why fermented

feed improves meat quality. The explanation in some studies is that probiotics such as *Bacillus subtilis* and *Enterococcus faecalis* play a key role in the effect of fermented feed on meat quality, and the metabolites produced by microorganisms in the process of fermentation might be beneficial to improve meat quality, such as the conversion of fast-twitch fibers to slow-twitch fibers [51]. Meat quality regulation by fermented unconventional feeds is likely facilitated by gut microbiota modulation [70], leading to improved growth performance, enhanced antioxidant capacity [71], and improved meat quality [72]. In addition, it was reported that the effect of fermented feed on meat quality might be due to the effects of specific components in fermented products, such as the anti-inflammatory and antioxidant effects of phenolic compounds [73] or flavonoids in the product that might have free radical scavenging activity [41]. Nevertheless, research into the mechanisms by which fermentation influences meat quality remains an area with significant room for further exploration.

**Table 5.** Effects of Fermented Unconventional Feeds on the Meat Quality of Monogastric Animals.

| Substrate and Dosage | Animal and Stage | Meat Quality | Reference |
|---|---|---|---|
| Okara (55% substitution for corn, 72% substitution for soybean meal) | Growing pigs, 55 days | Longissimus thoracis muscle a* value significantly increased from 16.75 to 17.5 compared to the control group | [41] |
| *Ginkgo biloba* leaves (4.5 g/kg) | Broilers, 1–42 days | Breast muscle 24-h drip loss significantly decreased from 5.46% to 4.41% compared to the control group | [42] |
| Pomace-mixed silage | Finishing pigs, eight weeks | The back fat's polyunsaturated fatty acid significantly increased from 10.60% to 12.15% compared to the control group | [24] |
| Soybean hulls (15%) | Finishing pigs, four weeks | The cooked pork fragrance scores significantly increased from 4.79 to 5.18 | [74] |
| Herbal residues (5%) | Broilers, 1–42 days | The breast muscle's steaming loss significantly decreased from 0.33% to 0.28% compared to the control group | [43] |
| Malic acid (8 g/kg) | Broilers, 1–42 days | The breast muscle's dropping loss significantly decreased from 4.20% to 2.63% compared to the control group | [44] |
| Ginkgo biloba leaves (0.35% in the starter phase, 0.4% in the grower phase) | Broilers, a 42-day experiment | Cooking loss significantly decreased from 14.26% to 11.93% compared to the control group | [21] |
| Dandelion (500 mg/kg addition) | Broilers, 1–42 days | Drip loss decreased from 3.07% to 2.60% compared to the control group | [40] |

### 3.4. Intestinal Health

The intestinal microbiota is a pivotal component of animal physiology, with crucial roles in nutrition, immunity, and defense against pathogens [75]. Numerous meta-analyses have demonstrated that incorporating fermented plant-based feed ingredients could enhance growth performance and digestibility across several growth stages in pigs [76]. This positive effect could be attributed to the capacity of fermentation to bolster the animal's intestinal health and potentially serve as an alternative to antibiotics. Antibiotics could promote animal growth because they have a germicidal effect. Reducing the use of antibiotics could increase the incidence of diarrhea in weaned piglets and increase the mortality of broilers. The probiotics used in fermentation, such as *Bacillus* sp., could significantly counter this phenomenon by improving intestinal health and increasing the daily weight

gain of animals. Therefore, fermented unconventional feed could be a good replacement for antibiotics [77].

Extensive research supports the ability of fermented feeds to ameliorate intestinal health in animals. Fermented unconventional feeds have been the subject of extensive investigation in this regard, and some of their impacts are summarized in Table 6. Their impact on the intestinal microbiota is primarily manifested as alterations in the gut microbiota and its metabolic byproducts [78].

The distinctive properties of fermented unconventional feeds induce gut acidification and establish favorable conditions for the proliferation of beneficial bacteria. Fermentation also eliminates anti-nutritional factors like saponins and gossypol found in unconventional feeds, which could be detrimental to the intestinal tract [79]. Weaned piglets, transitioning from liquid to solid feed, often experience stress responses that could lead to diarrhea [80]. Fermentation plays a significant role in addressing this challenge. Liquid fermentation feeds have gained increasing attention from scholars and industry professionals as weaned piglets better adapt to liquid substances, effectively mitigating the issue of diarrhea [12]. Moreover, research has suggested that the mammalian intestinal microbiota exists even before birth, with the mother seeding the fetal gut microbiota [81]. Some studies have used high-throughput sequencing technology to confirm that intestinal microorganisms have colonized the intestines during the development of chicken embryos and that these microorganisms mainly came from the reproductive system of the hens [82,83]. Consequently, enhancing maternal intestinal health with fermented feeds could positively influence the gut health of the offspring and aid in alleviating weaning stress in piglets. Additionally, the metabolic products generated through fermentation, such as probiotics and organic acids, impact the animal's gut and regulate the central nervous system [84]. This regulation is achieved through activation of the gut-brain axis and includes critical neurotransmitters like serotonin, which play pivotal roles in the gut and the brain [85]. Aroma-rich fermented feeds could modulate the animal's mood and increase its feed intake [86]. Studies have shown that fermented feeds could elevate the animal's average daily gain while enhancing its gut microbial flora, providing substantial evidence of this phenomenon [25,32,87]. The process is illustrated in Figure 2.

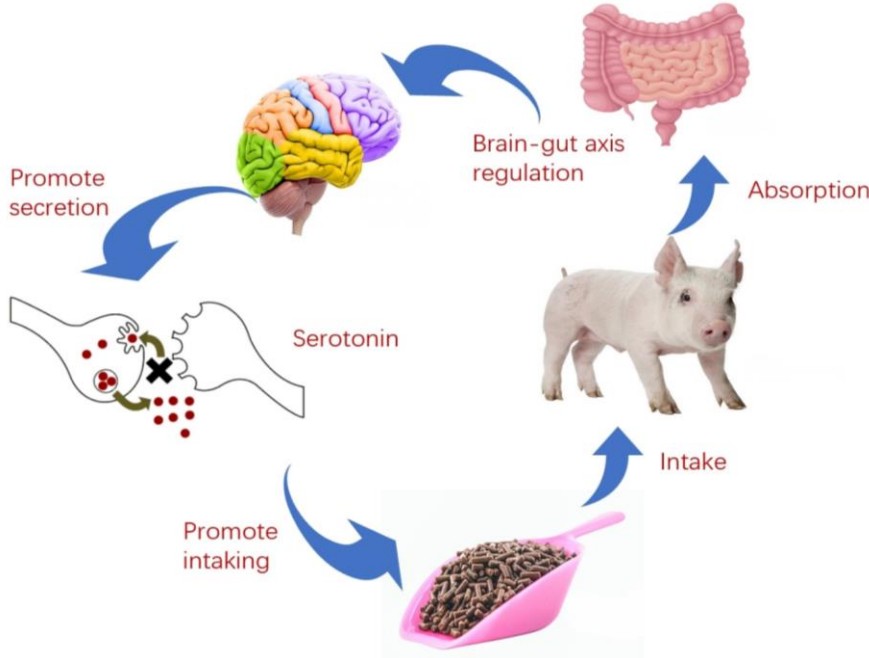

**Figure 2.** Brain-gut Axis Regulation of Fermented Feed Intake.

**Table 6.** Effects of Fermented Unconventional Feeds on Gut Health in Monogastric Animals.

| Substrate and Dosage | Animal and Stage | Gut Health | Reference |
|---|---|---|---|
| Cottonseed meal (80 g/kg) | Broilers, 1–42 days | Significant increase in the quantity of *Lactobacillus* in the cecum | [49] |
| Citri Sarcodactylis Fructus by-products (3%) | Broilers, 1–42 days | Significant reduction in the Shannon index of the cecum | [45] |
| Tea residue (3%) | Laying hens, 34 weeks old, a 6-week experiment | Significant decrease in the quantity of *Faecalibacterium* in the cecum | [88] |
| Heat-treated rice bran (5%) | Laying hens, 20 weeks old, an 8-week experiment | Significant increase in the relative abundances of *Lachnospira* and *Clostridium* | [89] |
| Dandelion (1000 mg/kg addition) | Broilers, 1–21 days | Significant reduction in the Shannon index of the cecum | [40] |
| Herbal residues (5%) | Broilers,1–42 days | Significant reduction in the indices ofcecum Chao1, Simpson, and Shannon indices | [43] |

## 4. Challenges and Prospects

While fermented feeds offer numerous advantages, they also face several challenges, including: (1) Complex microbial communities: Issues arise during production due to the complexity and diversity of microbial strains active in the fermentation process. These include contamination by unwanted microorganisms, transfer of antibiotic resistance genes, production of toxic metabolites, and induced immune overactivity [90]. (2) Strain source and safety: Ensuring the standardization of microbial strain sources and their safety and specific application characteristics is a paramount concern when preparing fermented unconventional feeds [91]. (3) Mechanisms of nutritional improvement: The mechanisms behind nutritional value enhancement during fermentation need to be better understood, necessitating further research. (4) Mixed microbial fermentation: Data on mixed microbial fermentation processes, especially the identification of dominant strains within the mixed culture, are lacking. Given the varying abilities of different strains to degrade anti-nutritional factors in unconventional feeds, they should be screened for strains that efficiently degrade these components. (5) Pelleted vs. non-pelleted feeds: The decision to produce pelleted or non-pelleted feed in the fermentation process requires further investigation. Non-pelleted feed might yield better results, as beneficial substances in fermented feeds, such as live probiotics, may be lost during the pelleting process. Pelleted feeds could improve the gelatinization degree of feed starch, enhance enzyme activity, passivate the anti-nutritional factors, and denature protein, all beneficial to the digestion and absorption of livestock and poultry. Therefore, choosing between them is difficult. (6) Single-strain vs. multi-strain fermentation: While single-strain fermented feeds have been widely researched in China, research on multi-strain fermented feeds is lacking. Exploring the development of multi-strain feed fermentation, particularly the discovery of dominant strains within these mixtures, holds promise. (7) Segmented fermentation: Establishing segmented fermentation, e.g., aerobic and anaerobic stages, is an emerging trend. (8) The economic benefit of fermented unconventional feed needs to be evaluated. On the one hand, fermented feed could improve feed conversion rate, shorten the time to reach the target body weight, and effectively use agricultural waste by turning it into treasure with a certain economic value; however, fermented feed entails equipment costs. Improper preservation could lead to feed deterioration and increase the associated costs.

Fermented feed prospects include: (1) Feeding with liquid fermented feeds: Adopting liquid feeding after biological fermentation as a feed formulation method could represent a significant trend in transforming and upgrading the livestock feed industry. However, this approach is still in its early stages in China, with limited commercial applications due

to its associated high costs. Widespread adoption might require more time and promotion. (2) Nutritional database for fermented feeds: As the production and use of fermented feeds increase, establishing a comprehensive nutritional database for these feeds becomes essential. This is particularly urgent given the potential benefits fermented feeds have for animal health and environmental sustainability. Developing a scientific assessment system is also crucial. (3) Exploration of alternative forms: While research in China has predominantly focused on fermenting single unconventional feed ingredients, alternative forms such as fermented concentrates and other products warrant increased attention from feed enterprises and research institutions.

## 5. Conclusions

As unconventional feed resources are important, we should develop and utilize them well. Due to the significant advantages of fermentation, unconventional feeds have great application prospects in feeding monogastric animals. This article reviewed the research progress of fermented unconventional feeds, from classification to technology and its application in monogastric animals. Additionally, the main challenges that restrict the development of fermented unconventional feeds were summarized, and the future trends were prospected. This review provides a theoretical basis for the future development of the feed industry.

**Author Contributions:** H.S.: conceptualization. H.S., X.K. and D.C.: literature search. H.S., X.K. and H.C.: writing—original draft preparation. H.T. and H.C.: writing—review and editing. D.C. and H.T.: funding acquisition. All authors contributed to the article and approved the submitted version. All authors have read and agreed to the published version of the manuscript.

**Funding:** This work was supported by the Special Project of Rural Revitalization Strategy in 2023-Agricultural Science and Technology Development and Resources and Environmental Protection Management Project (grant number 2023KJ115).

**Institutional Review Board Statement:** Not applicable.

**Data Availability Statement:** Data are available from corresponding authors upon request.

**Conflicts of Interest:** As the first author of the review, I declare that, Dan Chen, Huize Tan, Haoxuan Sun, and myself are employees of Wen's Food Group Co., Ltd, which is located in Yunfu City, China. The authors declare that the research was conducted in Wen's Food Group Co., Ltd.

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
