# Peer review of "Progress in Fermented Unconventional Feed Application in Monogastric Animal Production in China"

_fermentation, doi:10.3390/fermentation9110947_

Round 1
Reviewer 1 Report
Comments and Suggestions for Authors#1
Pigs are monogastric animals or animals that have a simple stomach. Their digestive system is relatively simple and they don’t have the ability to digest and utilize large amounts of fibrous material in their diet like ruminants do.
There are three main categories of feed resources and non conventional feed resources. Corn and soybean have been the major components of conventional feed for monogastric and satisfy the needs for energy and protein respectively.
However, these components of feeds have become very expensive and in some cases such as corn, scarce. Scientists have been looking for ways to find alternative feed sources for these two components.
Such non conventional products can be obtained from baking, grain milling, egg processing, brewing, distilling and refining industries. Before any non conventional feed resource is added diet there are a few considerations that must be analyzed and tested:
Choices in Selection of non conventional feed resources.
Types of con conventional feed resources.
Plant non conventional feed resources.
Factors affecting Inclusion rate of non conventional feed resources.
Non conventional feed resource are very useful in the production of pork. Hence a wide variety of experiments have been carried out to find the sources with the best nutritional values in terms of lysine and energy contents. Feed costs are very high in the monogastric industry and alternative feed sources are keys to making the production systems a profitable one both in the economic and financial departments.
#2
The article presented for review deals with a very interesting topic regarding alternative sources of feed for monogastric animals produced mainly by lactic fermentation. As a reviewer, I do not see any editorial errors. However, I believe that the work is a review of an area, e.g. China. The authors focused their attention mainly on this region of the world.
The abstract and introduction are written correctly. The arrangement of the chapters is correct. I consider the cited literature and its selection good.
Tables and figures are legible and described correctly.
As a reviewer, I believe that the Discussion section should be more extensive, but it is not very necessary.
I believe that the article can be published without corrections.

Reviewer 2 Report
Comments and Suggestions for Authors
L25 L25-28 is not a good beginning, and I cannot see any connection between this passage and feed antibiotics.
L31-32Under aerobic conditions, it cannot be called fermentation. Fermentation is defined as the process where electrons are transferred to an organic molecule serving as the ultimate electron acceptor, which only occurs under anaerobic conditions.
L32-44This section tracing the history of fermented feed is particularly suitable for presentation in a science history textbook, but it serves no purpose in this article. The entire paragraph is pointless and the author is just going around in circles without even touching on the topic of this article.
L45-67 The author is so focused on China's feed policy that, to be honest, such a specific paper may not be suitable for publication in an international journal, and it may make more sense to publish in the region. Our readership is not the same, and articles without broad interest are not recommended for publication in this journal
L77-86 I'm extremely disappointed to read this passage, as it contains no new information and is completely copied from textbooks in the 1980s.
L90 We cannot take the reported individual cases as consensus. In fact, currently, the microorganisms relied on for fermentation are only lactic acid bacteria and yeast, neither of which has the ability to degrade fibers. Therefore, after fermentation, water-soluble carbohydrates and protein degradation may occur, while the content of cellulose remains unchanged or increases slightly.
I have no interest in reading further. From the beginning of the paper, the article does not provide enough new knowledge, and the content discussed in the paper cannot arouse the wide interest of readers. Such a large copy of the textbook is simply unthinkable, and I do not think the article is suitable for publication in this journal
Comments on the Quality of English LanguageThe lauguage is fine.
Reviewer 3 Report
Comments and Suggestions for Authors
This review discusses the progress of fermentation of unconventional feed in monogastric animals. It deals with an important area of animal feeding. However, some sections should be improved.
Comments:
There are no supporting references for many parts. Please revise the whole manuscript.
Does the method of fermentation affect the final product? Discuss.
Table 2: are Ginkgo biloba leaves energy feed?
L128-130: not all silage feeds are suitable for monogastric animals. The final product differs according to the feed used. Ensilage is primarily used for high moisture forages and roughages for ruminant feeding. Please support this part with studies done on monogastric animals and their outcomes.
L180-183: please explain this result. Why?
3.2. Immune Function: this part needs more explanation of how fermented feeds improve the immune system.
3.3 Animal Products: please support this part with studies evaluating the effect of feeding fermented unconventional feed on the taste and odor of meat. And explain the effects on meat quality.
Figure 2. Brain-gut axis regulation of fermented feed. This figure doesn’t illustrate the brain-gut regulation at all. The authors should replace it with detailed information about this regulation or delete it.
4. Discussion: this title is unsuitable. Change to challenges and prospects
The authors should mention the economic aspects of feed fermentation.
Comments on the Quality of English LanguageFine
Reviewer 4 Report
Comments and Suggestions for Authors
General consideration
The topic is interesting. There is growing interest in the exploitation of agro-industrial wastes to produce fermented, protein-rich animal feed within the livestock industry, nevertheless, the review does not fulfill its purpose.
The reported literature is very limited, and not extensive.
Most of the references (> 70%) are from Chinese authors. Does not the rest of the world work on this topic?
Differences between various species should be commented on. The results come from fish as well as pigs without attribution.
Present the results from the same paper in different tables. If you cite a paper on pigs using a certain fermentation technique, you should have the results on the animal growth.
The discussion is poor.
It should be about the results presented in the previous section:
Some of the challenges have not been introduced previously (see pelleted as well as segmented fermentation)
Introduction
Should be less general and more focused on the topic.
Is the topic related only to China? Try to broaden the discussion
Line 26-30: it is not clear the relationship between antibiotic reduction and the use of fermented feed.
Line 49-51: expand the references
Line 56-59: the comparison with ruminants is not pertinent. They are not part of the subject
Line 69-71: report the classification, even in the supplementary material
Line 152: the species must be specified (In common carp or geese …)
Table 3- 4-5 are not easy to read
Line 168-174: avoid repetition
Line 178: add ref
Line 180: add ref
Line 234-240: the reported refs are about pigs only, while the table reports also broilers
Line 249: explain why they can serve as an alternative to antibiotics
Line 261-63: avoid repetition
Line 267: the reference refers top lamb which are not monogastric
Line 275: this happens in geese. Could you provide the same results for other species?
Discussion
Line 310: what is the situation in other parts of the world?
Round 2
Reviewer 3 Report
Comments and Suggestions for Authors
No further comments. Thank you